# The impact of climate change on ecology of tick associated with tick-borne diseases

**Heejin Choi, Chang Hyeong Lee** [ID]*

Department of Mathematical Sciences, Ulsan National Institute of Science and Technology, Ulsan, Republic of Korea

* chlee@unist.ac.kr

## Abstract

Infectious diseases have caused significant economic and human losses worldwide. Growing concerns exist regarding climate change potentially exacerbating the spread of these diseases, particularly those transmitted by vectors such as ticks and mosquitoes. Tick-borne diseases, such as Severe Fever with Thrombocytopenia Syndrome (SFTS), can be particularly detrimental to elderly and immunocompromised individuals. This study utilizes a mathematical modeling approach to predict changes in tick populations under climate change scenarios, incorporating tick ecology and climate-sensitive parameters. Sensitivity analysis is performed to investigate the factors influencing tick population dynamics. The study further explores effective tick control strategies and their cost-effectiveness in the context of climate change. The findings indicate that the efficacy of tick population reduction varies greatly depending on the timing of control measure implementation and the effectiveness of the control strategies exhibits a strong dependence on the duration of implementation. Furthermore, as climate change intensifies, tick populations are projected to increase, leading to a rise in control costs and SFTS cases. In light of these findings, identifying and implementing appropriate control measures to manage tick populations under climate change will be increasingly crucial.

## Author summary

This study investigates the impact of climate change on tick populations and the transmission of Severe Fever with Thrombocytopenia Syndrome (SFTS) in Korea using a mathematical model. We project a significant increase in tick abundance under future climate scenarios, emphasizing the need for effective control measures. Our model evaluates the efficacy of mowing and acaricide spraying in mitigating tick populations and associated SFTS cases. We find that both methods are effective when implemented for sufficient durations, with acaricide spraying showing greater impact, particularly under intensified climate change. Importantly, we demonstrate that the timing and duration of control implementation significantly influence their effectiveness. While mowing presents an environmentally friendly alternative, it requires prolonged implementation

**Data availability statement:** The data used in this study are available from https://github.com/choikijin/Climate_Jeju.

**Funding:** This work was supported by a National Research Foundation of Korea (NRF)

grant funded by the Korea government (MSIT) (2022R1F1A1064487) and by the BK21 Program (Next Generation Education Program for Mathematical Sciences, 4299990414089), funded by the Ministry of Education (MOE, Korea) and National Research Foundation of Korea (NRF). HC was supported by Basic Science Research Program through the National Research Foundation of Korea (NRF) funded by the Ministry of Education (RS-2024-00405375). The funders had no role in study design, data collection and analysis, decision to publish, or preparation of the manuscript.

**Competing interests:** The authors have declared that no competing interests exist.

(over two months) to achieve substantial reductions in SFTS cases and associated costs. This research provides valuable insights for developing and implementing tick control strategies to mitigate the public health risks associated with climate change.

## Introduction

The COVID-19 pandemic has served as a stark reminder of the devastating impact of infectious diseases on global health and economies. While significant efforts have been dedicated to addressing this immediate threat, concerns are mounting about the long-term implications of climate change on the spread of infectious diseases. Climate change is expected to alter environmental conditions, potentially expanding the geographical ranges of disease vectors and influencing their life cycles. This poses a particular threat for vector-borne diseases, which are already prevalent in many regions worldwide.

Since 1940, numerous emerging infectious diseases have manifested globally, with vector-borne diseases constituting 22.8% of these occurrences [1]. Within the spectrum of vector-borne diseases posing a threat to human health, those transmitted through the bite of infected ticks, such as Lyme disease, Crimean-Congo hemorrhagic fever, tick-borne encephalitis, and severe fever with thrombocytopenia syndrome (SFTS), are notable. SFTS, prevalent predominantly in East Asia, is an infectious disease classified as a zoonosis, capable of transmission between humans and animals. SFTS is caused by the SFTS virus, a member of the *Phlebovirus* genus within the *Bunyaviridae* family [2–4]. The broad host range of ticks, encompassing wild animals, humans, livestock, and companion animals, facilitates the transmission of the SFTS virus, leading to high seroprevalence rates in various animal species [5–7]. The primary clinical manifestations of SFTS include fever, anorexia, nausea, thrombocytopenia, and leukopenia [4,8–10]. In Korea, the inaugural confirmed case surfaced in 2013, with subsequent confirmed cases showing a steady increase. As of 2022, the cumulative case fatality rate of SFTS in Korea stands at 18.7% [11], emphasizing its significant risk factor compounded by the absence of a vaccine or antiviral treatment. Hence, mitigating the risk of SFTS entails avoiding exposure to ticks, underscoring the importance of studies of tick ecology to control SFTS infection effectively.

Given the diverse and intricate life cycles of ticks across different species, numerous investigations have been undertaken to comprehend their ecological dynamics. Studies by Ogden et al., Wu et al., Cheng et al., and Wu and Zhang [12–15] employed mathematical models to elucidate the behavior of the vector of Lyme disease, *Ixodes scapularis*, in Canada. Similarly, Dobson et al. [16] devised a process-based population model utilizing empirical data to delineate the ecology of *Ixodes ricinus*, a crucial vector in both medical and veterinary contexts in Europe. Wang et al. [17] introduced an agent-based model focusing on *Amblyomma americanum*, a tick species inhabiting the south-central United States. Furthermore, Kada et al. [18] explored the significance of stage-dependent factors in the tick dispersal model, examining the influence of the two principal tick families, hard ticks and soft ticks.

Climate change stands as a significant global challenge, with direct and indirect ramifications on tick ecology. Extensive research has been conducted to predict the impact of anthropogenic activities such as fossil fuel use and urbanization on future climate conditions [19–21]. The Intergovernmental Panel on Climate Change's Sixth Assessment Report (IPCC AR6) introduced a new framework in 2022 that combines Shared Socioeconomic Pathways (SSP) scenarios and Representative Concentration Pathways (RCP) scenarios. Depending on the climate change scenario and geographical region, average temperatures are projected to increase

by 1 °C to 7 °C or more by 2100, accompanied by substantial variations in humidity-related factors [22–25]. Climate change can directly change the behaviors of ticks, including the pre-oviposition period, survival duration, questing period, and host-seeking behavior [26–29]. For example, Li et al. [30] employed a mechanistic, agent-based model to predict the impact of climate warming on tick population ecology, suggesting heightened tick-host encounters and an increased risk of Lyme disease. Similarly, Nah et al. [31] demonstrated that alterations in certain tick characteristics due to climate change may elevate the risk of tick-borne encephalitis in the future. Moreover, climate change can influence the geographical distribution of tick and host habitats, as well as the length of the tick season [32–36]. Gilbert found that tick abundance is positively related to warmer climates and host abundance, suggesting that ticks could become more abundant in high altitudes due to global warming [37]. Alk-ishe et al. [38] observed that under the Representative Concentration Pathway (RCP) scenario, new habitats for *Ixodes ricinus* are projected to emerge across Europe. Elzinga et al. [35] utilized a mathematical model to forecast the stability of tick and host populations amid shifts in seasonal lengths prompted by climate change, alongside an exploration of harvesting strategies for moose population control.

In Korea, the predominant vector species implicated in Severe Fever with Thrombocytopenia Syndrome (SFTS) is *Haemaphysalis longicornis*, accounting for approximately 97.3% of ticks collected in the region [39]. While other indigenous Korean species like *Haemaphysalis flava*, *Haemaphysalis japonica*, *Ixodes nipponensis*, and *Amblyomma testudinarium* are recognized as SFTS vectors, their prevalence is considerably lower [39]. Consequently, this study primarily focused on *Haemaphysalis longicornis* due to its overwhelming dominance in Korea.

This study presents the development of a mathematical model to predict the population dynamics of *Haemaphysalis longicornis* on Jeju Island which is a subtropical island located at the southern tip of South Korea, providing a good habitat for ticks. The model includes four developmental stages reflecting tick growth, with each stage's rate of progression estimated based on climate factors such as temperature and humidity. The tick species under investigation belong to the *Ixodidae* family and exhibit a three-host life cycle. This life cycle involves three separate blood-feeding stages: larvae, nymphs, and adults. Consequently, the development rates and reproduction rate are estimated for each stage and incorporate climate factors. Temperature ($T$) and relative humidity ($H$) are chosen as the primary climate factors due to their influence on tick growth and habitat suitability. Previous studies have highlighted the importance of ambient temperature in predicting the development of *Haemaphysalis longicornis* and the incidence of SFTS cases [40,41]. Empirical experiments have demonstrated that an increase in temperature results in a shorter development period and a higher development rate [26,42,43]. On the other hand, Yano et al. [44] found that increasing humidity leads to a higher hatchability rate and a shorter developmental period for eggs. However, the study found no significant influence of relative humidity on the developmental period of other tick developmental stages. Based on this synthesis of previous studies, we establish climate-dependent development rate and reproduction rate parameters. Using mathematical methods, we estimate the parameters related to the climate-sensitive tick ecology based on collected tick data [45], and we perform the sensitivity analysis to assess the influence of the parameters on tick population dynamics. Furthermore, within the context of Shared Socioeconomic Pathways (SSP) scenarios, we investigate the impact of the duration and timing of control measure implementation on tick population dynamics and associated costs. Based on these findings, we discuss potential strategies for efficient tick control.

## Materials and methods

### Study area and data analysis

Jeju Island is located in the transition zone between the subtropical and temperate climate zones, and it maintains a warm and humid climate throughout the year, with average annual temperatures ranging between 16 °C and 17 °C and average annual humidity between 70% and 80%. According to [42,44], higher temperature and humidity accelerate tick growth and development rate. Consequently, Jeju Island's environment presents favorable conditions for tick habitation in Korea. Considering these epidemiological and environmental factors, Jeju Island is chosen as the primary study area for this research. Fig 1(a) depicts the SFTS incidence rate per 100,000 population in South Korea and Jeju Island. From 2017 to 2021, Jeju Island consistently exhibited a significantly higher SFTS incidence rate compared to mainland Korea. Notably, the incidence rate in Jeju Island in 2017 was about six times higher than that of Korea. Fig 1(b) shows the monthly average temperature and relative humidity in Jeju Island. The tick population data was collected on Jeju Island from April to November each year, spanning the period from 2017 to 2021. Tick collection was conducted utilizing dry ice bait-collecting traps, a specialized collection method, strategically deployed in four distinct environments characterized by high potential for human-tick interaction [39,46]. The collected tick data is presented in Fig 1(c). The data reveals a consistent trend, with a higher collection rate of nymph stages observed from April to July, and a higher collection rate of larval stages observed from August to November.

### Mathematical model

This study presents a stage-structured mathematical model, constructed using ordinary differential equations, to characterize the tick ecology on Jeju Island. The tick population is divided into four distinct development stages: $E$ denotes eggs, $L$ denotes larvae, $N$ denotes nymphs, $A^F$, and $A^M$ denote female and male adults, respectively. The model assumes that only female adults could lay eggs and subsequently perish.

The development rate from the egg stage to the larval stage is denoted by $d_1(T,H)$, where $T$ represents temperature and $H$ represents relative humidity. Development rates for subsequent stages are expressed as $d_i(T)$, $i = 2,3,4$, respectively. In particular, we defined the $d_4(T)$ as the reproduction rate of female adults. Furthermore, while larval development in many tick species is influenced by both overall tick density and mate-finding probability [18], the parthenogenetic nature of *Haemaphysalis longicornis* [47,48] allows us to simplify the model. We therefore focus solely on the density-dependent growth constraint using the carrying capacity term $K$. The parameter $K$ is determined by calculating the product of host density (hosts per square meter $m^2$) and the maximum number of ticks that can attach to a single host, separately for the grass and wood areas of Jeju Island. These two area-specific carrying capacities are then summed to yield the overall $K$ value [49]. Population growth is restricted by the total number $N_V$ of ticks capable of feeding on a host, which is represented by $N_V = L + N + A^F + A^M$. The natural mortality rate of ticks, $\mu_i$, $i = 1,2,\cdots,5$, is assigned a stage-specific value. Since the tick population data collected in the target area is only available on a monthly basis, the time unit of this model and the subsequent simulations in this model is set to one month. Additionally, we refer to research that studies the family *Ixodidae*, which includes *Haemaphysalis longicornis* and is commonly known as the hard ticks to set several parameters. A detailed estimation process of climate-dependent parameters can be found in S1 Text. The model equation is written as Eqs (1), and a schematic diagram of the model is shown in Fig 2, and the parameters used in our model are listed in Table 1.

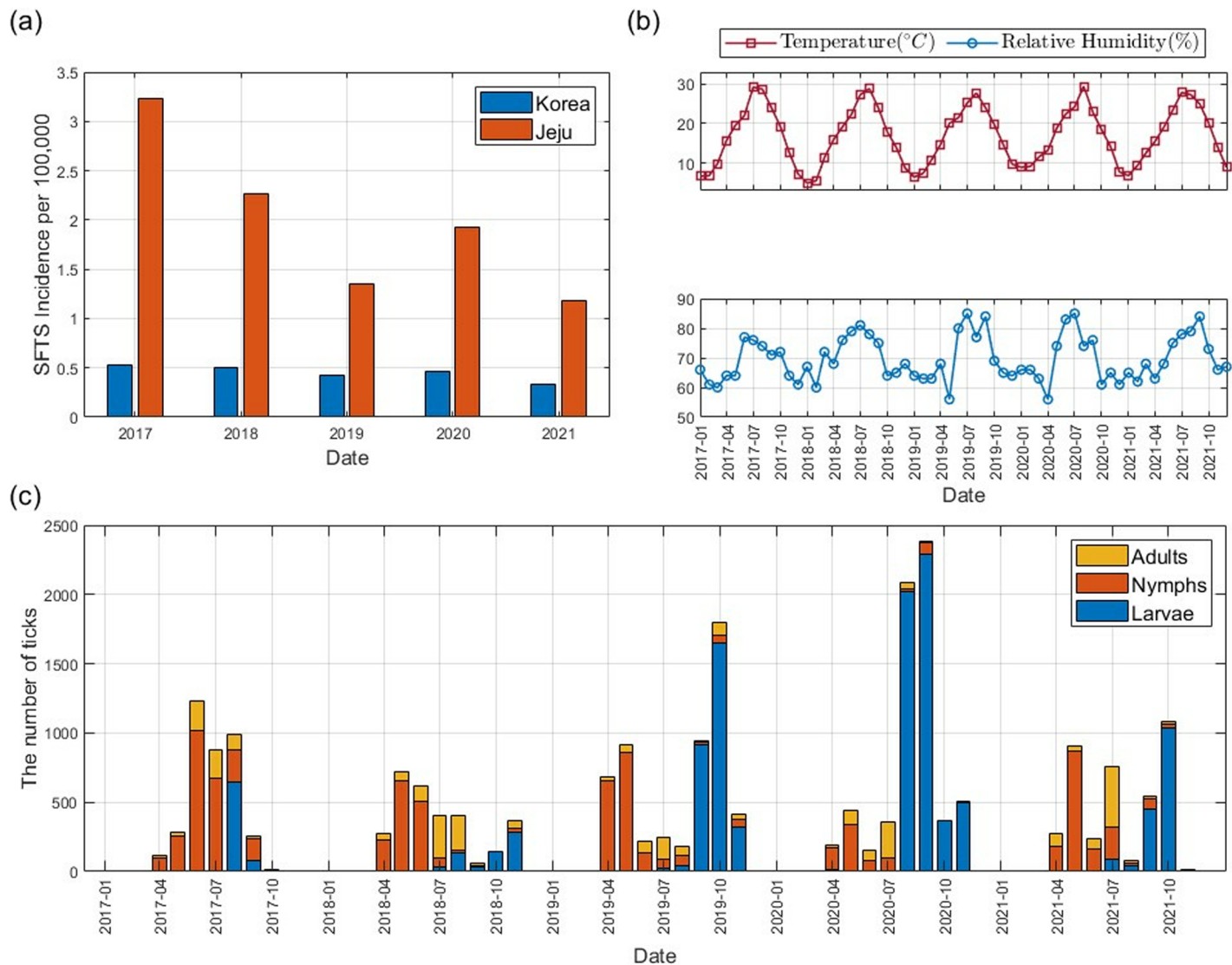

**Fig 1. Data collected in Jeju Island between 2017 and 2021.** (a) SFTS incidence per 100,000 population in South Korea and Jeju Island (b) monthly average temperature and relative humidity in Jeju Island (c) monthly number of ticks collected in Jeju Island [45].

$$\frac{dE}{dt} = d_4(T)\rho A^F - (d_1(T,H) + \mu_1)E$$

$$\frac{dL}{dt} = d_1(T,H)(1 - N_V/K)E - (d_2(T) + \mu_2)L$$

$$\frac{dN}{dt} = d_2(T)L - (d_3(T) + \mu_3)N \qquad (1)$$

$$\frac{dA^F}{dt} = \alpha d_3(T)N - (d_4(T) + \mu_4)A^F$$

$$\frac{dA^M}{dt} = (1 - \alpha)d_3(T)N - \mu_5 A^M$$

where $N_V = L + N + A^F + A^M$ and $K = M(A_W K_W + A_G K_G)$.

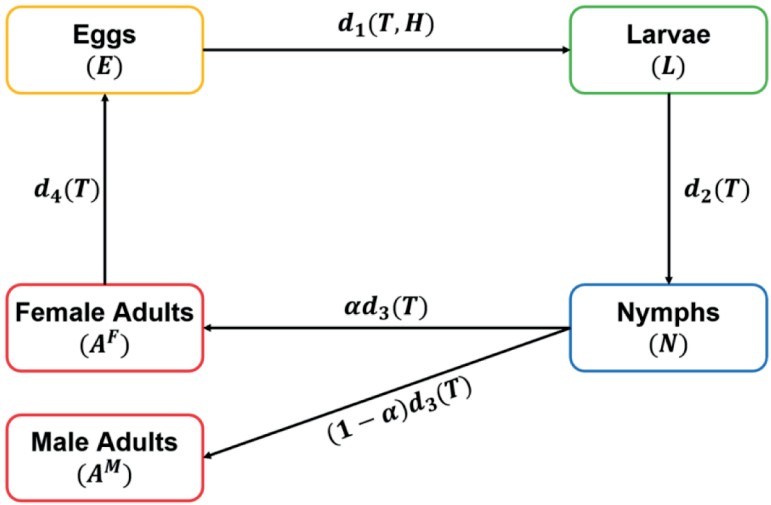

**Fig 2. Schematic diagram for the mathematical model.**

**Table 1. Parameter description.**

| Parameter | Definition | Value | Reference |
|---|---|---|---|
| $T$ | The temperature by month | vary | [50] |
| $H$ | The relative humidity by month | vary | [50] |
| $d_1(T,H)$ $= \max\{0, a_0 T^2 + a_1 H^2 + a_2 T \cdot H + a_3 T + a_4 H + a_5\}$ | The development rate from eggs to larvae depends on the temperature($T$) and relative humidity($H$) | Estimated | [42,43] |
| $d_2(T)$ $= \max\{0, a_6 T^2 + a_7 T + a_8\}$ | The development rate from larvae to nymphs depends on the temperature($T$) | Estimated | [42,43] |
| $d_3(T)$ $= \max\{0, a_9 T^2 + a_{10} T + a_{11}\}$ | The development rate from nymphs to adults depends on the temperature($T$) | Estimated | [42,43] |
| $d_4(T)$ $= \max\{0, a_{12} T^2 + a_{13} T + a_{14}\}$ | The reproduction rate of female adults depends on the temperature($T$) | Estimated | [42] |
| $\rho$ | The number of eggs laid per capita adult females | 3000 | [39] |
| $\mu_1$ | The mortality rate of eggs per month | 0.06 | [12] |
| $\mu_2$ | The mortality rate of larvae per month | 0.079 | [18] |
| $\mu_3$ | The mortality rate of nymphs per month | 0.025 | [18] |
| $\mu_4$ | The mortality rate of female adults per month | 0.03 | [18] |
| $\mu_5$ | The mortality rate of male adults per month | 0.03 | [18] |
| $K$ | Carrying capacity | 370000000 | Calculated |
| $M$ | Maximum number of ticks per host | 200 | [49] |
| $A_W$ | Woods area in Jeju Island ($m^2$) | 847180000 | [51] |
| $K_W$ | Carrying capacity for host per $m^2$ in wood area | 0.002 | [49] |
| $A_G$ | Grass area in Jeju Island ($m^2$) | 154550000 | [52] |
| $K_G$ | Carrying capacity for host per $m^2$ in grass area | 0.002 | [49] |
| $\alpha$ | Sex ratio of adults | 0.5 | [12,16] |

## Parameter estimation

We estimate the development rate and reproduction rate of ticks depending on the climate, $d_1(T,H)$ and $d_i(T)$, $i = 2, 3, 4$, using tick data presented in Fig 1(c). We fit each climate-dependent parameter into the quadratic function dependent on both temperature $T$ and relative humidity $H$ for $d_1$, or solely on temperature for $d_i$, $i = 2, 3, 4$. The least squares method is used to estimate these development rates and reproduction rate from the data. Tick data collection occurred annually between April and November, corresponding to the active period

for ticks. Consequently, tick data for December to March of the subsequent year was assumed to be zero. The estimation results are shown in Fig 3. Fig 3(a) compares the collected and estimated cumulative abundance of the total tick population including larvae, nymphs, and adults, and Fig 3(b) depicts the dependence of the estimated development rate and reproduction rate functions on climatic factors. As shown in Fig 3(b), all development rates and reproduction rate exhibit an increasing trend with rising temperatures. For $d_1(T, H)$, which incorporates both temperature and relative humidity, the development rate from eggs to larvae exhibits a positive correlation with both factors. However, the influence of temperature is observed to be greater than that of relative humidity. The coefficients of each climate-dependent parameter are in S1 Table and a comparison of the collected and estimated abundance of the tick population is in Fig A in S1 Text.

## Climate change scenarios

The SSP scenarios represent a combined framework encompassing two key challenges: mitigation and adaptation. Mitigation challenges include socioeconomic factors that influence reference emissions, while adaptation challenges encompass socioeconomic conditions, excluding physical climate change factors, that can influence a society's ability to adapt to a changing climate [53]. On the other hand, the RCP scenarios depict different levels of radiative forcing by 2100 [21]. Utilizing a scenario matrix that integrates SSPs and RCPs to depict radiative forcing at the end of the century, O'Neill et al. [54] further categorized these scenarios into two Tiers, with Tier 1 comprising four key scenarios for climate science research, SSP1-2.6, SSP2-4.5, SSP3-7.0 and SSP5-8.5. The Korea Meteorological Administration has provided detailed climate change projections for Korea based on these scenarios [50]. Fig 4 illustrates

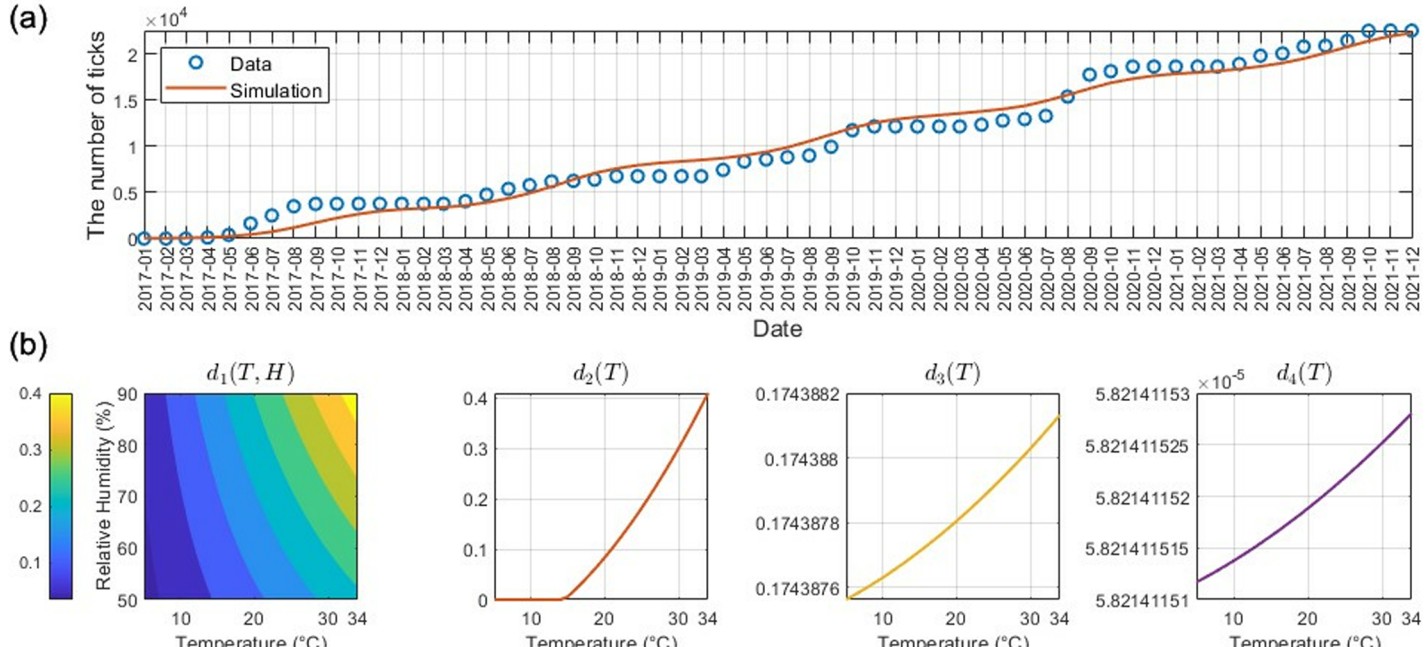

**Fig 3. Parameter estimation results.** (a) collected actual data (blue circle) and estimation result (red line) of the cumulative number of the total tick population including larvae, nymphs, and adults (b) estimation result of climate-dependent development rates and reproduction rate.

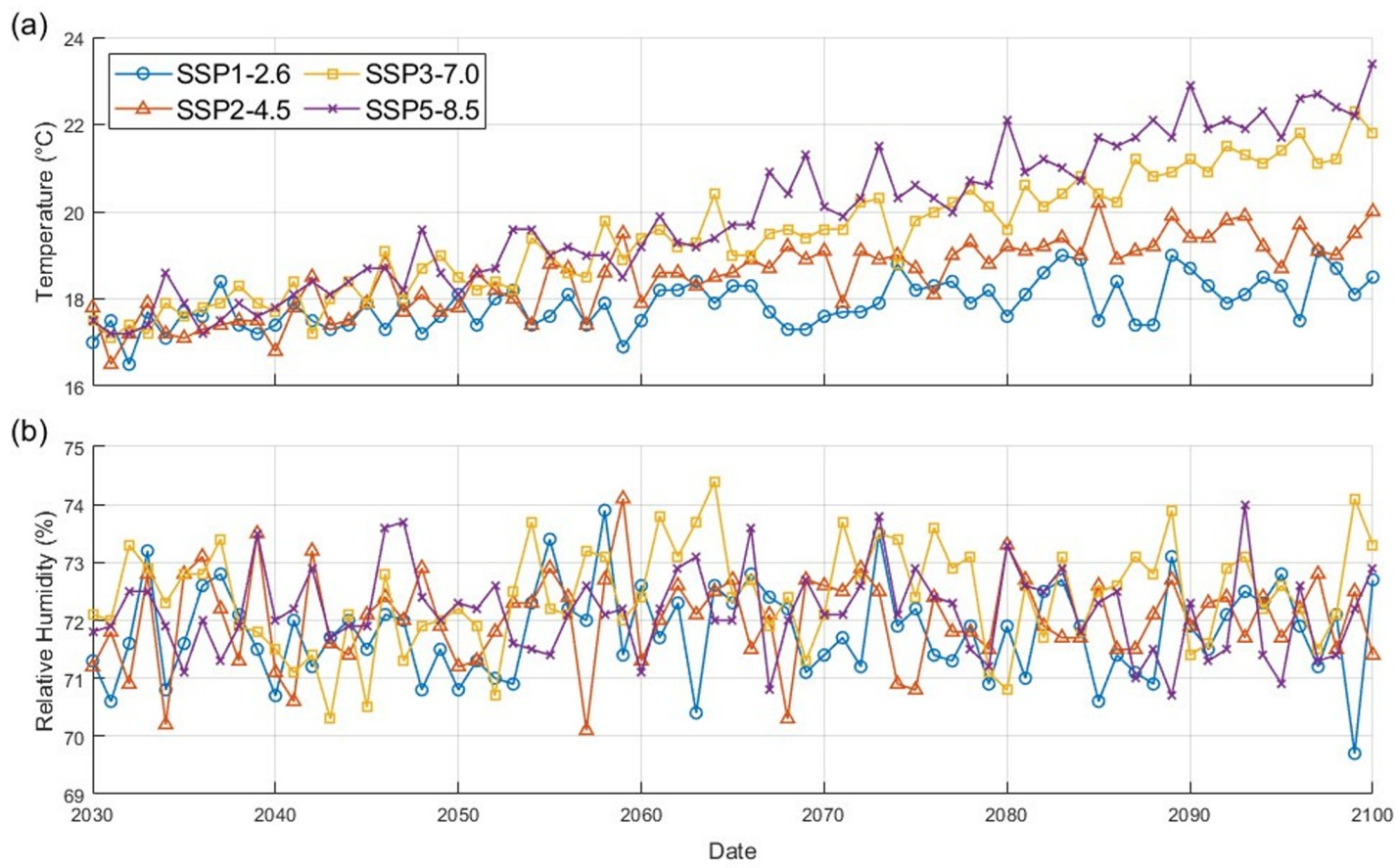

**Fig 4. Temperature and humidity on Jeju Island under SSP climate change scenarios.** (a) the average yearly temperature and (b) the average yearly relative humidity [50].

the annual average temperature and relative humidity for Jeju Island between 2030 and 2100 under the climate change scenarios. The annual average temperature shows minimal variation between the different scenarios until the 2060s, after which significant differences between SSP scenarios emerge, with a disparity of nearly 5 °C by 2100. Conversely, the relative humidity exhibits minimal variation across the SSP scenarios until 2100.

To further understand the factors influencing tick population dynamics, we perform a sensitivity analysis on climate variables $T, H$, and other parameters in Table 1. The normalized forward sensitivity index, denoted as $CI_p^{SSP}$, quantifies the sensitivity of the cumulative tick incidence under each SSP scenario to variations in the investigated parameter $p$. It is mathematically defined as follows [55]:

$$CI_p^{SSP} = \frac{\partial CI^{SSP}}{\partial p} \times \frac{p}{CI^{SSP}} \qquad (2)$$

## Control measures

Given the absence of a vaccine or antiviral treatment for SFTS, preventative measures are crucial for human health. These measures focus on avoiding tick bites and controlling tick populations. Current tick control strategies in Korea primarily rely on mowing and spraying acaricide. Previous studies [56,57] indicate that mowing can disrupt tick habitats and limit their access to hosts, thereby impeding their ability to feed. Based on this ecological principle, we incorporated a reduced tick development rate into our model to reflect the impact of diminished feeding opportunities. In contrast, the application of acaricides directly elevates tick mortality rates. Jang et al. [58] investigated the effect of mowing by comparing tick populations before and after mowing, over a period ranging from 2 to 19 days. Based on their findings, we fit the effect of one-month mowing to an exponential function, defining this fitted mowing effect as a control parameter $CR_1(t)$. This parameter is incorporated into our model as follows: $d_i^C = (1 - CR_1(t)) \times d_i$, for $i = 1, 2, 3$. Lee et al. [59] demonstrated that the efficacy of spraying acaricide against nymphs of *Haemaphysalis longicornis* under natural conditions diminishes over time. Similar to the mowing effect, we fit an exponential function to represent the effect of one-month acaricide spraying, defining this fitted effect as the control parameter, $CR_2(t)$. Additionally, Kim et al. [60] reported that the larvicidal efficacy of the acaricide was 57 times and 730 times higher than its efficacy against nymphs and adults, respectively. To account for these differences in stage-specific acaricide efficacy, we incorporate a constant factor to adjust the mortality rates in the model as follows: $\mu_2^C(t) = (1 + 57 \times CR_2(t)) \times \mu_2$, $\mu_3^C(t) = (1 + CR_2(t)) \times \mu_3$, and $\mu_4^C(t) = \mu_5^C(t) = (1 + \frac{57}{730} \times CR_2(t)) \times \mu_4$.

Implementing continuous mowing and acaricide spraying throughout the year may pose environmental and economic challenges. To address this, we establish various control scenarios based on the type of control measure and the duration of implementation per year. The control measure types are categorized as {Mowing, Spraying acaricide, Both} and the duration of implementations per year, {$1M, 2M, 3M, 4M$}. Here Both means both mowing and spraying acaricide are implemented, and $1M, 2M, 3M$, and $4M$ denote that the duration of implementation is one-, two-, three-, and four-month, respectively. During simulations, for scenarios involving mowing only, $CR_2(t)$ is set to zero and the mortality rates remain unchanged, $\mu_i^C = \mu_i, i = 2, 3, 4, 5$. Conversely, for scenarios involving only acaricide spraying, the mowing control parameter $CR_1(t)$ is set to zero, and the development rates are set to $d_i^C = d_i$ for $i = 1, 2, 3$. The model equation with the control measures implementation is written as Eq (3). The results of fitting $CR_1$ and $CR_2$ are presented in Fig B in S1 Text.

$$\frac{dE}{dt} = d_4(T)\rho A^F - (d_1^C(T, H) + \mu_1)E$$
$$\frac{dL}{dt} = d_1^C(T, H)(1 - N_V/K)E - (d_2^C(T) + \mu_2^C)L$$
$$\frac{dN}{dt} = d_2^C(T)L - (d_3^C(T) + \mu_3^C)N \qquad (3)$$
$$\frac{dA^F}{dt} = \alpha d_3^C(T)N - (d_4(T) + \mu_4^C)A^F$$
$$\frac{dA^M}{dt} = (1 - \alpha)d_3^C(T)N - \mu_5^C A^M$$

## Estimation of cost

We estimate the total cost associated with the tick population in two steps. Firstly, we calculate the total number of SFTS patients, $N_{SFTS}$ from 2030 to 2100 using the predicted monthly

tick abundance data $NT_i^j$ categorized by development stage $j$, where $j$ denotes larvae, nymphs, or adults. While ticks have different preferred host sizes depending on their developmental stage, they have the potential to bite humans at all stages except the egg stage. The biting rate is known to be influenced by ambient temperature. Gilbert et al. and Nielebeck et al. [29,61] observed that the proportion of ticks actively questing approaches 100% at 15 °C and this behavior persists as temperatures increase. To incorporate this temperature dependency into our model, we introduce $C_\eta(T_i)$, where $T_i$ is the temperature at time $i$ in a monthly unit, as a temperature-dependent parameter that adjusts the biting rate. To estimate the number of SFTS patients, we incorporate the monthly SFTS virus infection rate $\beta_i$ of ticks based on findings from [62]. Taking these into account, we estimate the total number of SFTS patients by 2100 as follows:

$$N_{SFTS} = \sum_{i=1}^{852} \sum_{j=1}^{3} NT_i^j \times C_\eta(T_i) \times \eta^j \times \beta_i \qquad (4)$$

In above, the time index $i$ represents a total of 852 months spanning the 71-year period from 2030 to 2100.

Secondly, we estimate the total cost ($C_T$), which is categorized into three components: medical cost ($C_M$), wage loss ($C_W$), and control measures cost ($C_C$). The medical cost ($C_M$) represents the direct medical expenses incurred for treating SFTS patients, the wage loss ($C_W$) accounts for the lost income of individuals suffering from SFTS and the control measures cost ($C_C$) represents the expenditure associated with implementing tick control measures. To estimate the medical cost and wage loss, we categorize patients into age groups, 70 years and older (70+), 60-69 years (60-69), 40-59 years (40-59), and 20-39 years (20-39), based on the actual distribution for SFTS patients [11]. Patient severity (Mild, Severe, Death) is determined by the probability of severe symptoms ($p_s$). Among severe cases, the number of fatalities is calculated using the age-specific SFTS mortality rate ($\delta_k$), reported by [11]. The medical cost of SFTS patients with mild symptoms is calculated only for hospital admission, which is calculated by multiplying the hospitalization fee per day ($F_H$), the recovery period ($1/\gamma$), and the number of patients with mild symptoms ($P_M$). In the context of our study, the medical cost for SFTS patients with severe symptoms includes both hospital admission and the potential utilization of Therapeutic plasma exchange (TPE), which is a treatment option for patients experiencing rapid clinical deterioration in Korea [63,64]. Wage loss is calculated for each age group by multiplying the average daily income ($W_k$), the employment rate ($E_k$), the recovery period ($1/\gamma$), and the number of patients within the age group $k$ ($P_k$). The control measures cost is calculated based on the specific type of control measure and the duration of implementations per year. We assume that the control measures will target the Olle Trail, a popular walking trail distributed throughout Jeju Island. For the cost estimation of control measures, the area subject to control measures is calculated as the product of the total length of the Olle Trail and the effective width of a typical road in Korea, reflecting the likely lateral reach of control measures [65,66]. The total cost ($C_T$) is calculated as the sum of the medical cost $C_M$, wage cost $C_W$, and control measure cost $C_C$. The detailed computation formula for the total cost is provided in Table 2 and the description of parameters for cost estimation is given in Table 3. A detailed description of the age distribution of SFTS patients, and the results of fitting $C_\eta(T_i)$ are provided in Fig C in S1 Text.

**Table 2. Formulae for the cost estimation.**

| Cost factor | Formula |
|---|---|
| Medical cost ($C_M$) | $P_M \times F_H \times 1/\gamma + P_S \times (F_H \times 1/\gamma + F_T \times N_T)$ |
| Wage loss ($C_W$) | $\sum_{k=1}^{4} P_k \times W_k \times E_k \times 1/\gamma$ |
| Control measure cost ($C_C$) | $F_C^n \times m \times y$ |

**Table 3. Descriptions and values of parameters for cost estimation.**

| Parameter | Description | Value | Reference |
|---|---|---|---|
| $NT_i^j$ | The number of tick in stage $j$ at time $i$ <br> $j$ = Larvae, Nymphs, Adults | Estimated | - |
| $N_{SFTS}$ | The number of SFTS patients | Estimated | - |
| $P_M$ | The number of SFTS patients with mild symptoms | Estimated | - |
| $P_S$ | The number of SFTS patients with severe symptoms | Estimated | - |
| $P_k$ | The number of SFTS patients in age group $k$ | Estimated | - |
| $C\eta(T_i)$ | Adjustment parameter for the biting rate of ticks | Estimated | [29,61] |
| $\eta^j$ | The biting rate of ticks in stage $j$ <br> $j$ = Larvae, Nymphs, Adults | [0.02, 0.062, 0.062] | [67,68] |
| $\beta_i$ | The SFTS virus infection rate of ticks in $i$-th month | [9.6, 18.9, 17.9, 6.7, 19.1, 7.8, 15.1, 14.7, 15.2, 13.6, 7.1, 5.3] | [62] |
| $p_s$ | Probability of becoming severe cases | 0.32 | [63] |
| $\delta_k$ | The mortality rate of SFTS in age group $k$ | [0.2474, 0.2264, 0.1053, 0] | [11] |
| $1/\gamma$ | The recovery period of patients | 10 | [63] |
| $F_H$ | The hospitalization fee per day | 470($) | [69] |
| $F_T$ | The hospitalization fee per procedure | 800($) | [70] |
| $N_T$ | The average number of TPE procedure | 3 | [71,72] |
| $W_k$ | The average daily income in age group $k$ | [68.08, 82.14, 128.24, 93.015] | [73] |
| $E_k$ | The employment rate in age group $k$ | [0.23, 0.566, 0.757, 0.655] | [74] |
| $F_C^n$ | The cost per implementation of each control measure <br> $n$ = Mowing, Spraying acaricide, Both | [18000($), 21600($), 39600($)] | [56,75] |
| $m$ | The duration of control measure implementation per year | 1, 2, 3, 4 (month) | Assumed |
| $y$ | The simulation period (year) | 71 | Assumed |

## Results

### Effects of climate change

In this section, we investigate the impact of climate change on tick population dynamics. We employ SSP scenarios to simulate tick population changes from 2030 to 2100.

Fig 5(a) shows the monthly number of ticks under different climate change scenarios, and Fig 5(b) shows the cumulative number of ticks from 2030 to 2100. For the simulations, the initial tick population size for 2030 is set to $[E, L, N, A^F, A^M] = [875, 791, 303, 1078, 1076]$. This initial condition is derived from simulation results assuming constant climate conditions between 2021 and 2030. As shown in Fig 5(a), under the SSP1-2.6 scenario, a steady decline in the monthly abundance of tick population is observed throughout the simulation period. In contrast, SSP3-7.0 and SSP5-8.5 scenarios exhibit a decrease in the the monthly tick abundance until 2060s, followed by a significant increase thereafter. This divergence in tick population trends aligns with the substantial rise in average temperature projected for both scenarios after the 2060s, as depicted in Fig 4(a). These results suggest a strong influence of

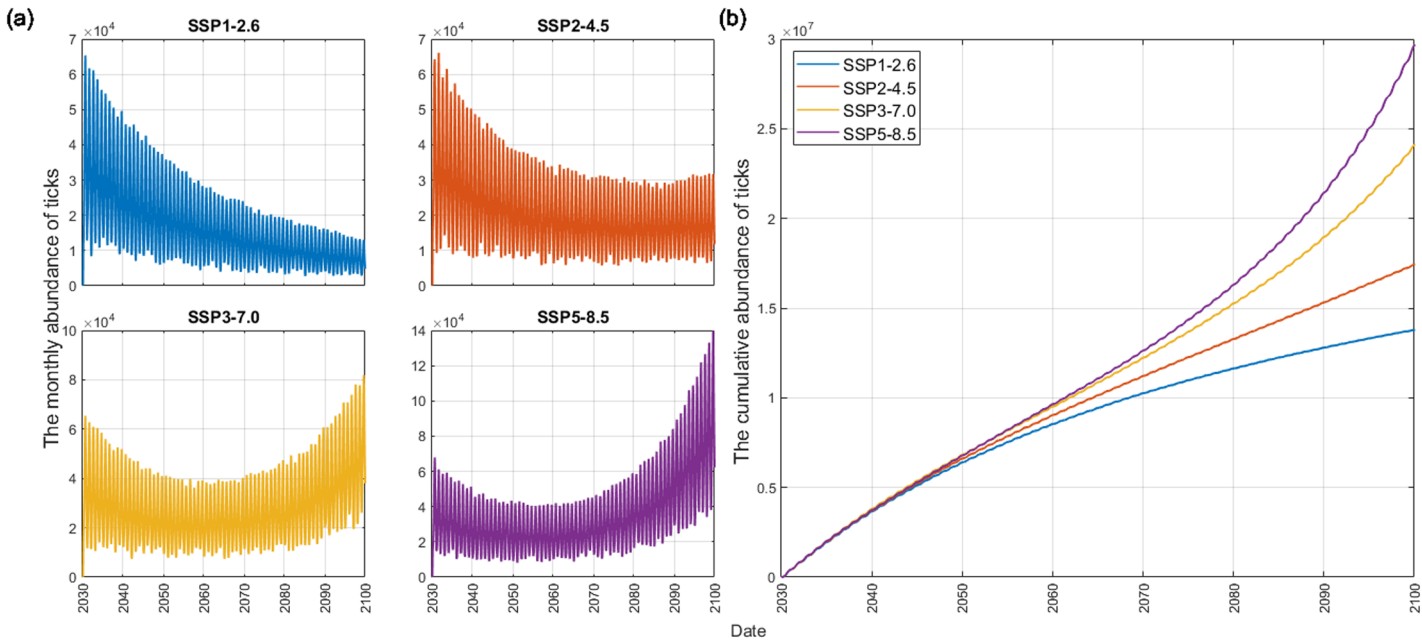

**Fig 5. Effects of climate change under SSP scenarios.** (a) the monthly abundance of ticks (b) the cumulative abundance of ticks from 2030 to 2100.

average temperature on tick abundance. Fig 5(b) depicts the substantial difference in cumulative tick abundance between SSP1-2.6 and SSP5-8.5 scenarios in 2100, with the latter reaching nearly double the value. Considering that the historical maximum monthly tick count from 2017 to 2021 in the target area was approximately 2500, these results suggest a potential for significantly higher annual tick populations in the future without effective climate change mitigation strategies or appropriate control measures. More information about setting the initial condition of simulation is provided in S2 Text.

For the sensitivity analysis, the 10000 sets from a uniform distribution within the range of $\pm 1\%$ are selected randomly for each climate factor and parameter. The cumulative tick abundance under the SSP scenario($CI^{SSP}$) is then simulated from 2030 to 2100. Fig 6 shows the sensitivity index of various parameters for the SSP scenarios. Overall, the impact on $CI^{SSP}$ is observed to be proportional to the severity of climate change. As inferred from the climate-dependent parameters in Fig 3 (b), the climate factors, the temperature($T$), and relative humidity($H$), are the positive influential parameters. *Additionally, the parameters $d_4$ and $\mu_4$ related to the reproduction and mortality of female adults exhibit the strongest influence on $CI^{SSP}$* and demonstrate the highest sensitivity to climate change. This suggests that the population size and fecundity of female adults significantly affect the tick population dynamics. On the other hand, the mortality rate of male adults is the least influential parameter, and the sensitivity index is almost zero.

### Effects of control measures

In this section, we evaluate the effect of tick control measures using a mathematical model. The initial conditions for the simulations are set to be same as in Fig 5 of the previous section. We assume control measures would be implemented annually from April to November, coinciding with the known period of tick activity.

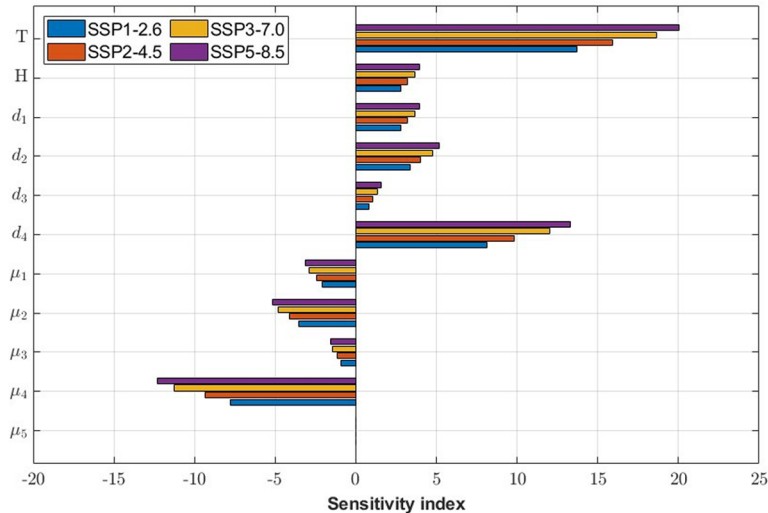

**Fig 6. Sensitivity index of the cumulative number of ticks about model parameters under the SSP scenarios.** $T$ denotes temperature, $H$ denotes relative humidity, $d_1$ denotes the development rate from eggs to larvae, $d_2$ denotes the development rate from larvae to nymphs, $d_3$ denotes the development rate from nymphs to adults, $d_4$ denotes the reproduction rate of female adults. Additionally, $\mu_1$ denotes the mortality rate of eggs per month, $\mu_2$ denotes the mortality rate of larvae per month, $\mu_3$ denotes the mortality rate of nymphs per month, $\mu_4$ denotes the mortality rate of female adults per month, and $\mu_5$ denotes the mortality rate of male adults per month.

Fig 7 shows the cumulative number of ticks by the year 2100 according to different SSP scenarios, various timing of control measure implementation, and three types of control measure for one-month(1M) and four-month(4M) scenarios respectively. The results indicate that implementing control measures during specific months can effectively reduce the tick population, even for interventions with the same intensity. Furthermore, mowing is less effective than acaricide spraying in reducing tick populations. As shown in Fig 7, if mowing is implemented as the sole control measure for one month, it should commence in July for all SSP scenarios except SSP3-7.0 to achieve optimal tick population reduction. Conversely, acaricide application is most effective when started in May under all SSP scenarios. Additionally, the most comprehensive intervention, applying Both control measures for four months per year, can lead to a cumulative tick abundance reduction to approximately 20,000, regardless of any SSP scenario. The cumulative number of ticks collected under each control scenario, including scenarios with no control measures implemented for durations of 1M, 2M, 3M, and 4M, are presented in S2 Table and S3 Table, respectively. The figures presenting results for two- and three-month control measures and the results for the reduction rate compared to when there is no control measure are displayed in S1 Fig and S2 Fig, respectively. Additionally, simulations including all possible control measure combinations based on the duration of interventions implemented per year are presented in S3 Fig–S5 Fig.

Fig 8 illustrates the annual tick abundance according to the best and worst scenarios for each control measure under the SSP5-8.5. For a given control measure type, the best and worst scenarios are determined by the timing of implementation. Fig 8 shows that the tick population dynamics can vary depending on the timing and duration of implemented control measures. Changes in dynamics according to the timing of control measures are more pronounced when only a single control measure is taken. Especially in the case of mowing, which is less effective when implemented alone, the gap between the best and worst scenarios

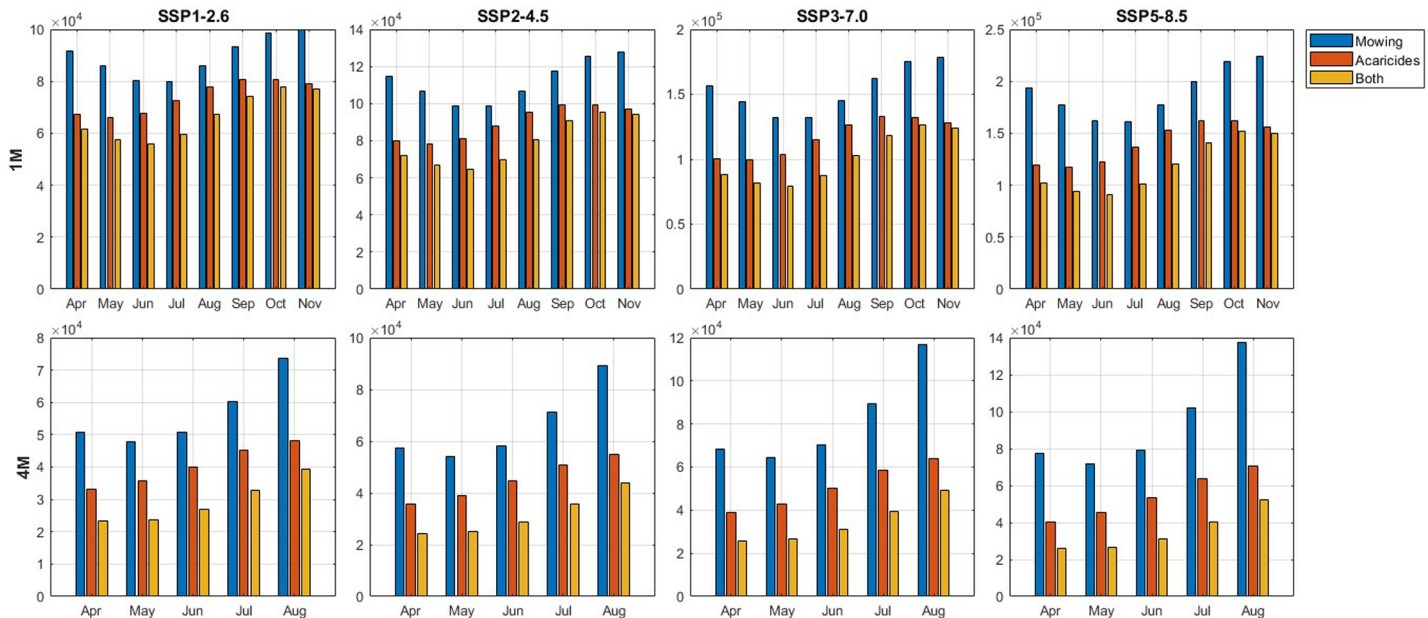

**Fig 7. Cumulative number of ticks when each control measure is implemented for 1M and 4M.** For the 4M scenario, each control measure is implemented for four consecutive months from the starting month to the following 3 months. For example, if the control measure is implemented starting in April, the control measure will be implemented in April, May, June, and July.

is substantial. Furthermore, while "Both(Best)" represents the most effective strategy in the one-month(1M) scenario for the tick population reduction, the population begins to increase as it approaches 2100 under the most severe climate change scenario (SSP5-8.5). However, implementing both control measures for more than two consecutive months annually leads to a steady decline in the tick population until 2100. The starting month in which each control measure is implemented when simulating the Best and Worst scenarios according to SSP scenarios is provided in S4 Table and the results of annual tick abundance under the best and worst scenarios for each control measure in other climate change scenarios are provided in S6 Fig.

## Estimation of cost

To evaluate the cost for control measures of SFTS under climate change scenarios, we identify the most effective scenarios for reducing the tick population based on the type and frequency of control measures. Subsequently, we estimate the number of SFTS patients and the associated costs.

Fig 9 illustrates the cumulative abundance of ticks, the cumulative incidence of SFTS patients, total costs, and medical costs for each combination of the SSP scenarios and control measures. Implementing control measures, regardless of the specific type or duration, effectively reduces the number of SFTS patients and associated deaths compared to scenarios without control measures under all climate change scenarios. Furthermore, excluding the scenario with only mowing implemented under SSP5-8.5 with a one-month duration, all other scenarios reduce the cumulative number of SFTS patients below 355, which is equivalent to an average of 5 patients per year. When implementing only one control measure, acaricide spraying reduces the number of SFTS patients more proportionately compared to the tick

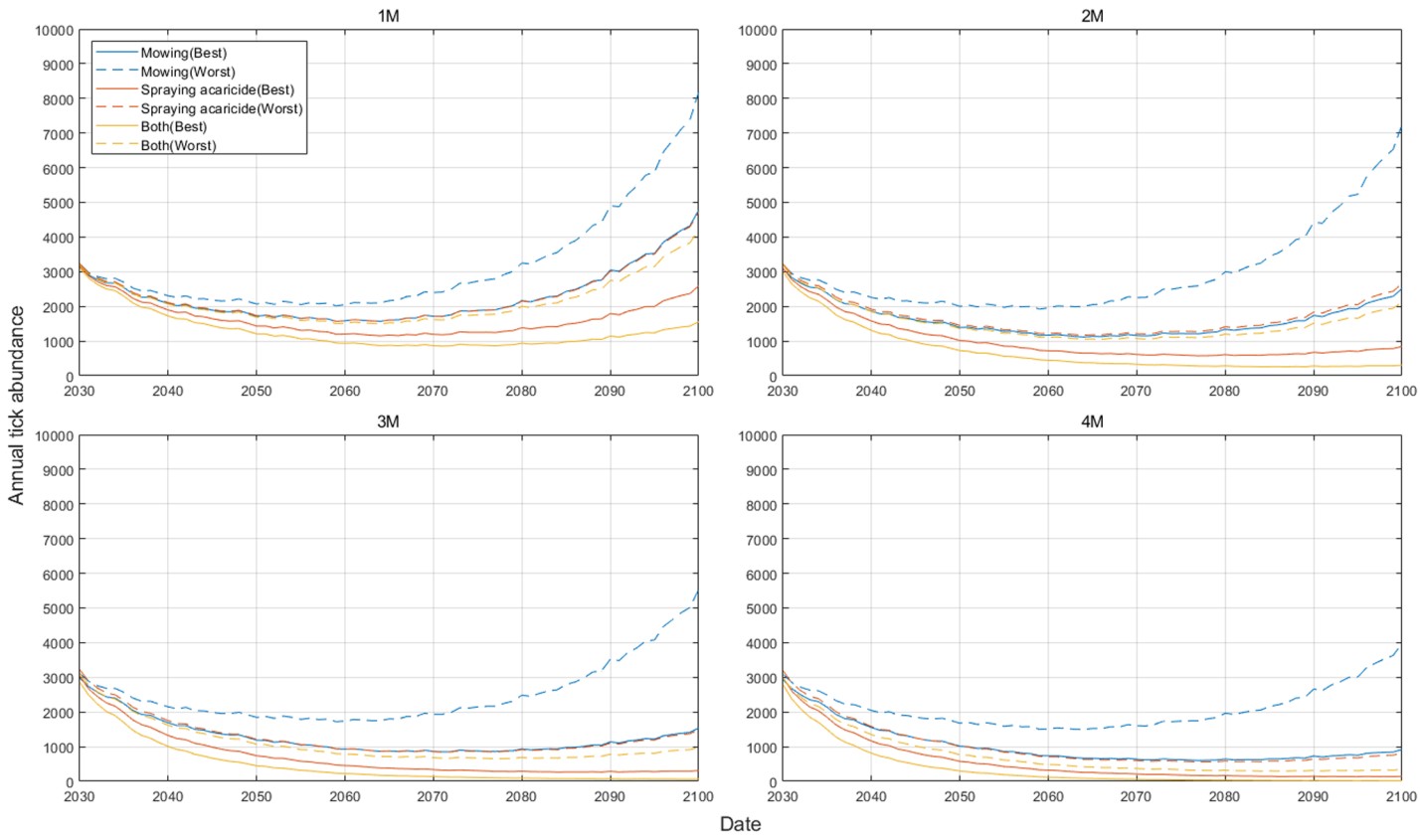

**Fig 8. The annual tick abundance under the best and worst scenarios for each control measure under the SSP5-8.5.** Best and Worst indicate the control measure scenario that reduces the tick population the most and the least, respectively.

population reduction resulted from mowing alone. In particular, under the SSP1-2.6 scenarios, acaricide spraying results in fewer SFTS patients despite incurring higher total costs. In contrast, under the SSP2-4.5, SSP3-7.0, and SSP5-8.5 scenarios, if a single control measure is implemented for one month, the difference in SFTS patients between the two control measure scenarios exceeds 50. Consequently, the associated medical costs outweigh the difference in implementation costs, leading to a higher total cost when only mowing is implemented for one month. Furthermore, implementing both control measures results in insignificant changes in the number of SFTS patients under all SSP scenarios, leading to limited variations in total cost. The annual incidence of SFTS patients under various climate change scenarios and control measures is displayed in S7 Fig–S10 Fig, along with illustrative examples of SFTS patients categorized by severity and age group are presented in S11 Fig–S14 Fig. Moreover, a more detailed cost analysis, in comparison to the no control scenario, is provided in S3 Text.

## Discussion

Climate change poses a significant threat to human health by altering the distribution and dynamics of vector-borne diseases [76,77]. Ticks, as vectors of various pathogens, are particularly sensitive to environmental fluctuations, making them a crucial focus for climate change impact assessments[78]. However, research on tick ecology associated with climate change, particularly in the context of Korea, remains relatively limited. This study aimed to address

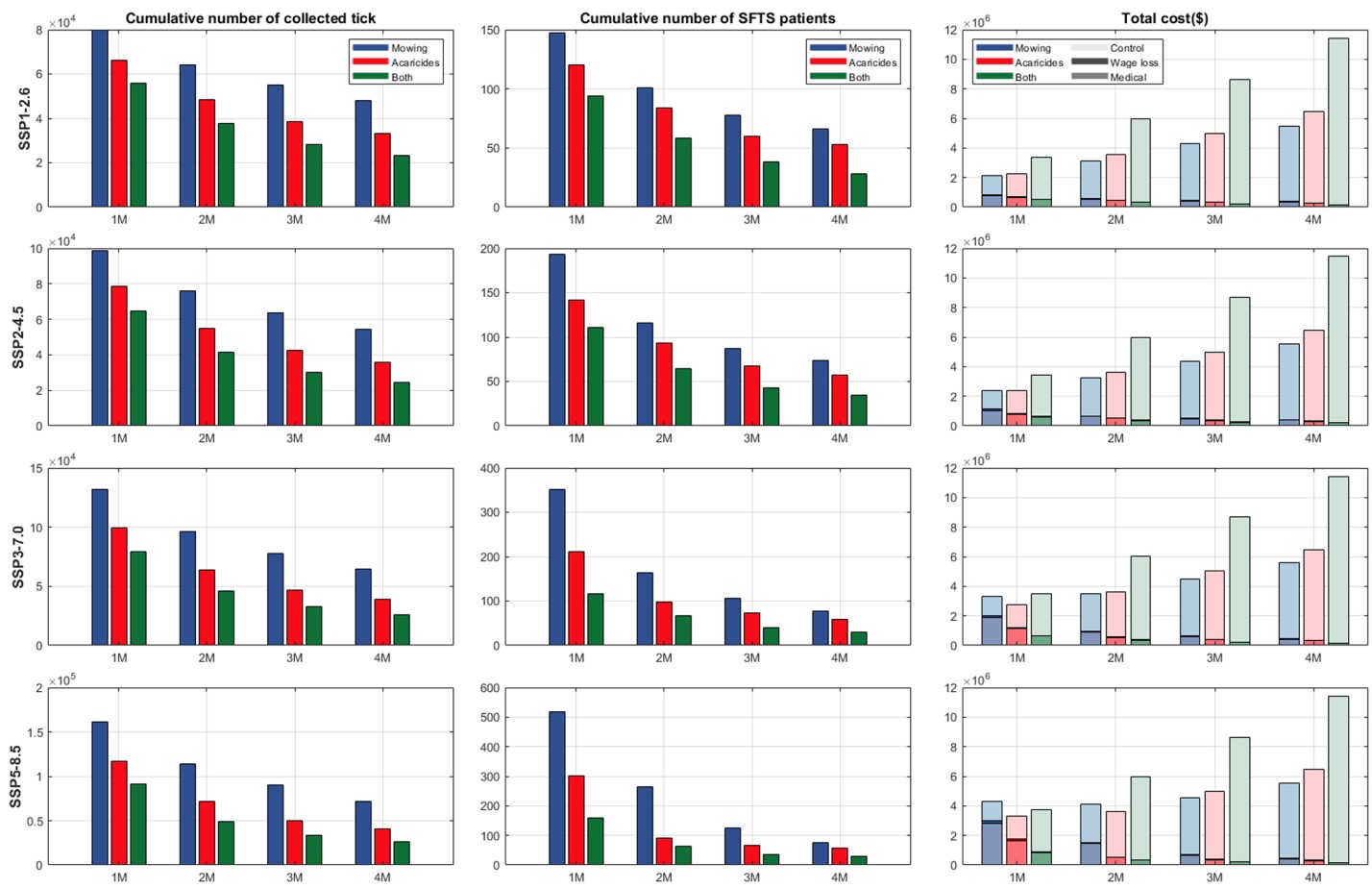

**Fig 9. The cumulative number of the tick, the cumulative number of SFTS patients, and total cost including medical cost, wage loss, and control cost according to different climate change scenarios and control measures.** In the right column (Total cost), in each color (blue, red, green), the lightest shade corresponds to control costs, the darkest shade represents wage losses, and the intermediate shade indicates medical costs.

this gap by developing a comprehensive mathematical model to investigate the impact of climate change on tick ecology and SFTS transmission on Jeju Island in Korea.

The developed model incorporated temperature and humidity as key climate factors influencing tick development. The model also considered the actual life cycle of ticks, including eggs, larvae, nymphs, and adults, and accounts for the distinct ecological characteristics of each stage. Rigorous mathematical methods were employed to estimate the parameters related to climate-sensitive tick ecology. The parameter estimation process involved utilizing actual data of tick population, climate projections, and biological knowledge to ensure the accuracy and reliability of the model.

The model was used to simulate the potential impacts of climate change on tick populations under the SSP climate change scenarios. Fig 5 indicated that climate change could lead to an increase in tick abundance, particularly under the climate change scenarios such as SSP3-7.0 and SSP5-8.5, which implies that the number of ticks may increase significantly, as global warming worsens. In addition, the sensitivity analysis (Fig 6) showed that the relationship between climate factors and the ecology of ticks is apparent. In addition, Yoon et al. [79] analyzed the statistical analysis of the tick population and climate factors in Korea and found

that the increase in monthly average temperature and relative humidity affected the increase in the tick population, which was consistent with the sensitivity analysis of our model to climate factors. Therefore, it is important to find appropriate control measures to control tick populations under climate change.

To mitigate the potential adverse effects of climate change, the study explored the effectiveness of different tick control strategies. The analysis considered two primary control methods in Korea: mowing and spraying acaricide. The model evaluated the impact of these control strategies under varying durations and combinations. It was assumed that tick control would be conducted from April to November, covering the period from spring to autumn in the Korean region, and the duration of the control ranges from one month to four months. Bickerton et al. [80] investigated the effectiveness of insecticide spraying to control *Haemaphysalis longicornis* in New Jersey, USA, through a field experiment. Although the most effective timing of insecticide spraying was different, they reported that the rate of tick reduction could vary depending on the timing of insecticide spraying, similar to our results, and that it was effective to spray in a row for several months to reduce the tick population.

One of this study's key findings is the critical importance of tick control measures to prevent the increase in tick populations and associated SFTS cases under climate change scenarios. Sensitivity analysis revealed that female adult tick mortality and reproduction rate exerted the most significant impact on tick abundance. This finding is attributed to the reproductive role of female adult ticks. Consequently, effective tick population control strategies should prioritize the management of female adult tick abundance.

Concerning the effectiveness of control strategies, our results demonstrated that both mowing and spraying acaricide can effectively reduce the tick population when implemented over a sufficient duration (at least two months), and the effectiveness of reducing the tick population varies greatly depending on the timing of control measure implementation. If tick control is conducted for the shortest duration of one month, the most effective month for spraying acaricide to reduce tick populations, under any climate change scenario, was found to be in May. For mowing, the most effective month was found to be in July, except under the SSP3-7.0 scenario, where June was more effective. When both policies are implemented together, the greatest reduction in tick populations was observed when conducted in June. Moreover, spraying acaricide exhibited a stronger impact on tick population reduction compared to mowing, particularly under scenarios with higher climate change intensity. In addition, the effectiveness of control strategies is highly dependent on the duration of implementation. Longer control periods lead to more substantial reductions in tick populations. Under the most severe climate change scenario (SSP5-8.5), a long-term strategy for tick abundance reduction should involve the implementation of both mowing and acaricide spraying for a minimum duration of two months.

Nevertheless, due to the expansive distribution and habitat range of ticks, coupled with the influence of environmental factors, Jang et al. [58] emphasized the necessity for more ecologically sound tick control strategies in place of acaricide spraying. Implementation of mowing, as an environmentally preferable alternative to acaricide spraying, resulted in a reduction in the tick population compared to the absence of any control measures. However, in accordance with the climate scenario, implementing mowing for only one month or two months resulted in a higher incidence of SFTS cases compared to other control measures, rendering it economically unfavorable. To achieve cost benefits from mowing as a standalone intervention, it is imperative to implement it for a duration exceeding two months. Consequently, this study proposes that a mowing regimen exceeding two months per year might offer a partial reduction in tick populations. This approach, if successful, could demonstrably decrease the number of SFTS cases while mitigating associated costs.

This study has certain limitations. First, data on tick populations in Korea is only available from 2017 to 2021. While we believe that this five-year dataset is sufficient for estimating parameter values related to tick ecology, a longer time series would be necessary for more precise parameter estimation. Even within this limited dataset, the number of collected larvae significantly exceeds that of other life stages, hindering accurate estimation of stage-specific populations using numerical approximation methods. Second, our model does not incorporate density-dependent effects on tick behavior. Tick behaviors such as survival and host-seeking can be influenced by both tick and host densities, as well as tick life stage [12,16,81]. However, due to the scarcity of studies on *Haemaphysalis longicornis* and to ensure model stability, we opted for constant parameter values. Third, our model assumes that control measures solely affect tick abundance without directly influencing SFTS infection parameters such as biting rates and transmission rates. While this simplification facilitates the analysis of control costs, it is important to acknowledge that this may not fully reflect the complexity of the effects of control measures. For instance, timely acaricide applications can suppress the host-seeking behavior of nymphs [82], and the effectiveness of acaricide in reducing tick-borne disease prevalence can vary depending on the specific pathogen [83]. These potential effects on tick behavior and disease transmission warrant further investigation and could be incorporated into future refinements of the model. Fourth, while we incorporate the influence of climate change on tick abundance, we acknowledge that climatic factors can also impact pathogen dynamics directly, as well as indirectly through effects on the density and behavior of reservoir hosts critical for pathogen maintenance. For instance, Ostfeld and Brunner highlighted the significant role of larval survival, influenced by climate, for certain tick-borne diseases [84]. Our current model does not explicitly account for these intricate host-pathogen interactions, focusing primarily on the broader relationship between climate change and tick population dynamics.

Despite these limitations, this study provides valuable insights into the potential impacts of climate change on tick populations, SFTS patient numbers, and associated costs by utilizing a mathematical model. The findings of this study can serve as a valuable foundation for future research investigating the complex interplay between climate change, tick population dynamics, and the economic burden of vector-borne diseases.

## Supporting information

**S1 Text. Estimation of parameters.**
(PDF)

**S1 Table. The formula of development rates and its coefficients.**
(PDF)

**S2 Text. Initial condition of simulation.**
(PDF)

**S2 Table. Cumulative number of the collected ticks in 2100 when each control measure is implemented 1M and 2M.**
(PDF)

**S3 Table. Cumulative abundance of ticks in 2100 when each control measure is implemented 3M and 4M.**
(PDF)

**S1 Fig Cumulative number of collected ticks when each control measure is implemented for 2M and 3M.**
(PDF)

**S2 Fig. Reduction rate of each control measure for different SSP scenarios.**
(PDF)

**S3 Fig. Simulation results including all possible control measure combinations based on the 2M scenario.**
(PDF)

**S4 Fig. Simulation results including all possible control measure combinations based on the 3M scenario.**
(PDF)

**S5 Fig. Simulation results including all possible control measure combinations based on the 4M scenario.**
(PDF)

**S4 Table. Starting month in which each control measure is implemented for best and worst scenario.**
(PDF)

**S6 Fig. The annual tick abundance according to the best and worst scenarios for each control measure.**
(PDF)

**S7 Fig. The annual tick abundance and SFTS incidence under SSP1-2.6 scenario.**
(PDF)

**S8 Fig. The annual tick abundance and SFTS incidence under SSP2-4.5 scenario.**
(PDF)

**S9 Fig. The annual tick abundance and SFTS incidence under SSP3-7.0 scenario.**
(PDF)

**S10 Fig. The annual tick abundance and SFTS incidence under SSP5-8.5 scenario.**
(PDF)

**S11 Fig. The plot of SFTS patients by severity under SSP1-2.6 scenario.**
(PDF)

**S12 Fig. The plot of SFTS patients by severity under SSP2-4.5 scenario.**
(PDF)

**S13 Fig. The plot of SFTS patients by severity under SSP3-7.0 scenario.**
(PDF)

**S14 Fig. The plot of SFTS patients by severity under SSP5-8.5 scenario.**
(PDF)

**S3 Text. Cost analysis.**
(PDF)

## Author contributions

**Conceptualization:** Heejin Choi, Chang Hyeong Lee.

**Data curation:** Heejin Choi.

**Formal analysis:** Heejin Choi, Chang Hyeong Lee.

**Funding acquisition:** Chang Hyeong Lee.

**Investigation:** Chang Hyeong Lee.

**Methodology:** Heejin Choi, Chang Hyeong Lee.

**Project administration:** Chang Hyeong Lee.

**Software:** Heejin Choi.

**Supervision:** Chang Hyeong Lee.

**Validation:** Heejin Choi, Chang Hyeong Lee.

**Visualization:** Heejin Choi.

**Writing – original draft:** Heejin Choi, Chang Hyeong Lee.

**Writing – review & editing:** Heejin Choi, Chang Hyeong Lee.

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
