## [Decision Letter · Decision Letter 0]

13 Jan 2025

PCOMPBIOL-D-24-01928

The impact of climate change on ecology of tick associated with tick-borne diseases

PLOS Computational Biology

Dear Dr. Lee,

Thank you for submitting your manuscript to PLOS Computational Biology. After careful consideration, we feel that it has merit but does not fully meet PLOS Computational Biology's publication criteria as it currently stands. Therefore, we invite you to submit a revised version of the manuscript that addresses the points raised during the review process.

Please submit your revised manuscript within 30 days Mar 15 2025 11:59PM. If you will need more time than this to complete your revisions, please reply to this message or contact the journal office at ploscompbiol@plos.org. Please include the following items when submitting your revised manuscript:

We look forward to receiving your revised manuscript.

Sincerely,

Narendra M Dixit

Academic Editor

PLOS Computational Biology

Thomas Leitner

Section Editor

PLOS Computational Biology

**Additional Editor Comments :**

The reviewers have appreciated the manuscript overall. They have, however, expressed concerns about some of the assumptions and modeling/data analysis details and made several other recommendations that would improve the manuscript. Please address all of their comments.

**Journal Requirements:**

2) Please amend your detailed Financial Disclosure statement. This is published with the article. It must therefore be completed in full sentences and contain the exact wording you wish to be published.

**Reviewers' comments:**

Reviewer's Responses to Questions

Reviewer #1: Overall I really like this paper, it does some interesting investigations and most comments are small editorial changes for clarity. However there are three things which are more fundamental and might mean issues with the model. These need addressing before publication, either by more justification in the text or they potentially require correction if I have understood what is happening correctly.

Page 4 line 128 you say you calculate K based on hosts per m2 for grass and wooded areas and then add them, but I don’t think that is what you should do. Surely you should do the average, or run the model on the two types of area, but adding them gives you a much bigger K than you would expect and will impact on all of your results going forward.

Page 5- the model : The functional forms of the development rates in your model come from fit to data I think, but did you consider making them mechanistic? For example you know that for eggs higher humidity leads to higher hatchability rate and lower development time so you could include that specifically but that does not seem to be D1. I don’t think what you have done is wrong but you should justify that choice when you have specific data that would allow you to do something mechanistic.

Parameter estimation, this bit seems key. You say you fit it using the data in figure 1c but I think that means each development rate is fitted to a different set of data ie larvae, nymphs and adults? So you could give a bit more detail here and explain that.

More potentially problematics is that the data is not the total number of ticks on the island, or even number per m2 which K is based on, so how do you take that into account in your parameter estimates? Surely is is only question ticks in a particular area? So you need to justify why fitting exactly to that data is the right thing to do. It might have been more appropriate to do the mechanistic modelling and make sure you have the right type of behaviour but not necessarily the right numerical behaviour.

Smaller comments

Page 2 line10 delete “year”

Lines 39 to 52 there is a new paper by Worton et al that you could include here (https://royalsocietypublishing.org/doi/full/10.1098/rsif.2024.0004) on modelling ticks with climate change. Also

Gilbert L, Aungier J, Tomkins JL . 2014 Climate of origin affects tick (Ixodes ricinus) host-seeking behaviour in response to temperature: implications for resilience to climate change Ecol. Evol. 4 , 1186–1198. (doi:10.1002/ece3.1014)

And Gilbert L . 2010 Altitudinal patterns of tick and host abundance: a potential role for climate change in regulating tick-borne diseases?. Oecologia 162 , 217–225. (doi:10.1007/s00442-009-1430-x)

Might be of interest for the biology.

Page 2 line 51 I don’t know what a Representative Concentration Pathway is, can you explain briefly?

Page 3 line 56 on: I think you should say that Haemaphysalis longicornis is in the Ixodes family and is the main vector of SFTS earlier than you do, so at the start of the paragraph which starts on page 56 which would make the rest of the paragraph make more sense.

Line 69 you say shorter development period and higher development rate, are these the same thing or are they more likely to develop ie have a lower death rate while developing? Can you make that clearer here please?

Line 78 again I am afraid I don’t know what Shared Socioeconomic pathways scenarios are, is this a standard method? It either needs a reference or more explanation.

Page 4 line 105 your say the data comes from four different environments, but that is not apparent from the graphs, are they all just added together? Do the different environments have different tick abundances? Ie do you use this info about the different environments in your model?

Line 116 delete the sentence that starts “additionally” you have already said this in the previous sentence.

Line 120 you refer to larval reproduction, but larvae don’t reproduce so what do you mean here?

Page 6 lines 160-164, if these had come earlier then that would have answered some of my previous questions.

Page 87 lne 196 delete “within”

Line 202 what do you mean by “one month acaricide spraying” . it could mean spraying monthly or just the effect only lasts for a month

Line 207 and 208 I am not sure why you have chosen to write the equations this way. What you have said in the text. I think you say that larve are 57 times more likely than nymphs and 730 times more likely than adults to die of the acaricide (line 204)

So I would say that means CR2*μ_2=730*CR2* μ_4=57*CR2*μ_3, that is sort of what is in the equations, but you have made the nymph death rate the “normal” one and I wondered why, can you at least confirm that the effect parameter was fitted to the nymph data? Perhaps we also need more information about CR2, is this the rate of application of acaricide or just the effect? If so how does the rate and size of application get included? I think you just have control on and off? Can you say that explicitly here

Page 10 line 296-300, presumably this is related to figure 3b where d1 and d2 have much bigger changes in response to temp and humidity than d3 and d4 where the y axis has an extremely narrow range?

Figure 7, I might add “see tables s2 and S3 to see the tick abundance with no control measures”- that was a question I had when I originally saw the graphs.

Figure 8; Did you consider plotting these all on the same scale on the y axis ie so the max value is 10000? It might be helpful for comparison, but if you have tried and it did not work for some reason then ignore this comment!

Page 11 line 354, I think this is because you are talking about the annual average rather than the cumulative number of patients, but this 5 does not come from the diagram so I found this confusing- can this be seen somewhere? If not perhaps say you “reduce the cumulative number of patients below x which is the equivalent to 5 per year.”

Figure 9 I don’t understand the right hand column, why are there faded and dark bits?

Page 13 line 425. I know this would be complicated to do all possible combinations, but given your results for the single controls did you look what happens if you mow in July and acaricide in May, is that better than doing both in June? I am not requiring you to do this, I was just interested.

Reviewer #2: Review of: The impact of climate change on ecology of tick associated with tick-borne diseases

The authors develop an ODE system to describe the dynamics of a tick population with reproduction and transitions depending on both temperature and humidity. The model is used to predict long term tick abundances, disease risk, and associated costs under different climate change scenarios and control strategies. The specific application considered in this paper is for Severe Fever with Thrombocytopenia Syndrome on Jeju Island in South Korea.

Model formulation:

The choice for density dependence needs further explanation/justification. Since d1 is the transition rate from eggs to adults, this term seems to imply that the hatching rate (either success in hatching and/or length of incubation) is density dependent. Do you have evidence for this? Moreover, this choice of nonlinearity does not appear to preserve non-negativity of the system since dL/dt |_L = 0 may be negative.

In addition, it is incorrect to say “larval reproduction” since larva do not reproduce (pg 4, lime 121).

In model (1) when defining AF and AM, use alpha and 1-alpha, respectively, for clarity. While your model is technically correct as written since you estimate alpha = ½, for a different estimate this would be incorrect.

Equation (4): For many hard tick species, adult male ticks do not take a bloodmeal. Is this the case here? Please clarify whether the total number of SFTS patients depends on total number of adults or just the number of female adults.

Related: Pg 9, lines 300-301: Is it almost zero or is it zero? If the adult males do not take bloodmeals, then I don’t see how males would impact this calculation at all.

Study site and control measures: Please include some measure of scale here. What is the spatial scale under consideration in this study? Is it the entirety of Jeju Island? And if yes, does, for example, mowing assume that mowing occurs over the entire area.

Results:

Figures: Some of the figures (e.g., Fig 5, 8) are missing axes labels.

Figure 5: Can you provide an explanation as to why in many of the scenarios we observe an initial decrease in the tick population?

Figure 7 caption: Remove “collected”. (I am assuming this is actual tick counts from the model, not a sampling estimate.)

Figure 9: I had difficulty reading the cost graph due to the resolution, but if I am reading this correctly, it looks like the control costs are significantly higher than the medical and wage loss costs. Given the information you have available, does this seem accurate?

Discussion: I think it may be worth pointing out in the discussion that, while the modeling of the tick population dynamics is detailed, the modeling of pathogen dynamics is very simplified. Specifically, disease risk is based on tick abundance and does not, for example, explicitly include dependence on competent hosts or pathogen dynamics. For example, it has been found that larva survival plays an important role in R0 for certain tick-borne diseases (e..g, Ostfeld Richard S. and Brunner Jesse L. 2015 Climate change and Ixodes tick-borne diseases of humans Phil. Trans. R. Soc. B37020140051).

Minor grammatical comments:

- Remove “the” before “elderly” in the abstract

- Pg 2, line 7: “range” � “ranges”

- Pg 2, line 10: remove “year”

- The models (equations (1) and (3)) should be given in a sentence

- Pg 13, line 417 add “the” between “reduce” and “tick”

**Have the authors made all data and (if applicable) computational code underlying the findings in their manuscript fully available?**

Reviewer #1: **No: **The data is there but I cannot see the code they used to run the models

Reviewer #2: Yes

PLOS authors have the option to publish the peer review history of their article (what does this mean?). If published, this will include your full peer review and any attached files.

Reviewer #1: **Yes: **Rachel Norman

Reviewer #2: No

**Figure resubmission:**
---

## [Decision Letter · Decision Letter 1]

24 Feb 2025

Dear Prof. Lee,

We are pleased to inform you that your manuscript 'The impact of climate change on ecology of tick associated with tick-borne diseases' has been provisionally accepted for publication in PLOS Computational Biology.

Best regards,

Narendra M Dixit

Academic Editor

PLOS Computational Biology

Thomas Leitner

Section Editor

PLOS Computational Biology

Reviewer's Responses to Questions

**Comments to the Authors:**

Reviewer #1: All comments have been taken on board, thank you

Reviewer #2: The authors have addressed all previous comments.

**Have the authors made all data and (if applicable) computational code underlying the findings in their manuscript fully available?**

Reviewer #1: Yes

Reviewer #2: None

PLOS authors have the option to publish the peer review history of their article (what does this mean?). If published, this will include your full peer review and any attached files.

Reviewer #1: No

Reviewer #2: No

---

## [Editor Report · Acceptance letter]

PCOMPBIOL-D-24-01928R1

The impact of climate change on ecology of tick associated with tick-borne diseases

Dear Dr Lee,

I am pleased to inform you that your manuscript has been formally accepted for publication in PLOS Computational Biology. Your manuscript is now with our production department and you will be notified of the publication date in due course.

With kind regards,

Anita Estes
